# Beyond Scattered Acceptance: Fast and Coherent Inference for DLMs via Longest Stable Prefixes

**Pengxiang Li**[1]   **Joey Tsai**[2]   **Hongwei Xue**[1]   **Kunyu Shi**[1]   **Shilin Yan**[1†‡]
[1]Accio Team, Alibaba Group   [2]Tsinghua University
[†] Project Leader   [‡] Corresponding Author

## Abstract

Diffusion Language Models (DLMs) promise highly parallel text generation, yet their practical inference speed is often bottlenecked by suboptimal decoding schedulers. Standard approaches rely on "scattered acceptance"—committing high-confidence tokens at disjoint positions throughout the sequence. This approach inadvertently fractures the Key-Value (KV) cache, destroys memory locality, and forces the model into costly, repeated repairs across unstable token boundaries. To resolve this, we present the **Longest Stable Prefix (LSP)** scheduler, a training-free and model-agnostic inference paradigm based on *monolithic prefix absorption*. In each denoising step, LSP evaluates token stability via a single forward pass, dynamically identifies a contiguous left-aligned block of stable predictions, and snaps its boundary to natural linguistic or structural delimiters before an atomic commitment. This prefix-first topology yields dual benefits: systemically, it converts fragmented KV cache updates into efficient, contiguous appends; algorithmically, it preserves bidirectional lookahead over a geometrically shrinking active suffix, drastically reducing token flip rates and denoiser calls. Extensive evaluations on LLaDA-8B and Dream-7B demonstrate that LSP accelerates inference by up to 3.4× across rigorous benchmarks—including mathematical reasoning, code generation, multilingual (CJK) tasks, and creative writing—while matching or slightly improving output quality. By fundamentally restructuring the commitment topology, LSP bridges the gap between the theoretical parallelism of DLMs and practical hardware efficiency.

## 1   Introduction

Diffusion Language Models (DLMs) have emerged as a compelling alternative to autoregressive generation, offering an intrinsically parallel inference process that leverages bidirectional context (Austin et al., 2021a). This paradigm holds the promise of significant latency reductions over traditional one-token-at-a-time decoding. However, this promise remains largely unfulfilled in practice. The iterative refinement process, central to DLM generation, is frequently bottlenecked not by the model's architecture, but by the *strategy* used to commit intermediate predictions. This creates a stark paradox: models designed for parallelism are often constrained by the sequential nature of their own convergence.

At the heart of this inefficiency lies the prevalent strategy of *scattered acceptance*, where tokens are committed independently based on local confidence (Nie et al., 2025) or in fixed-size, semi-autoregressive blocks (Arriola et al., 2025). This approach is fundamentally costly in two distinct ways. First, from an **algorithmic perspective**, it creates a fragmented sequence of frozen and mutable tokens. The numerous boundaries between these regions are unstable, requiring repeated, localized repairs that slow the convergence to a globally coherent output. Second, from a **systems perspective**, this fragmentation shatters the Key-Value (KV) cache into small, non-contiguous segments, destroying the memory locality that is critical for efficient Transformer inference. Consequently, the *active* (uncommitted) portion of the sequence remains long, keeping attention computationally expensive for many iterations.

In this work, we argue that overcoming this bottleneck requires a new commitment topology. We introduce the **Longest Stable Prefix** (LSP), a training-free, model-agnostic scheduling paradigm founded on the principle of *monolithic prefix absorption*. Instead of accepting scattered islands of confident tokens, LSP identifies and commits the longest contiguous, stable prefix of the remaining active sequence in a single atomic step. This is achieved through a lightweight, single-pass procedure: (1) it computes a logit margin score for each active position; (2) it adaptively selects a margin threshold to target a fractional block size (e.g., 25–50% of the active suffix); and (3) it snaps the candidate block's boundary to a nearby structural delimiter (e.g., punctuation or a newline) before committing. A simple fallback rule guarantees progress by committing at least one token per iteration, even when the model is highly uncertain.

This prefix-first geometry fundamentally alters the computational dynamics of DLM inference. By design, the frozen prefix grows as a single, contiguous block. This maximizes KV cache reuse and ensures subsequent attention queries are focused on a rapidly shrinking active suffix. The adaptive thresholding strategy encourages the active sequence length to decay geometrically, leading to a near-quadratic total work complexity that scales gracefully with sequence length. Algorithmically, committing structurally-aligned monolithic blocks minimizes the cross-boundary conflicts inherent in scattered acceptance, reducing the number of repair cycles needed to achieve a coherent state.

Our contributions are threefold:

- We identify scattered acceptance as a primary bottleneck in DLM inference and propose *monolithic prefix absorption* as a more efficient commitment topology. We instantiate this principle in LSP, a novel, training-free scheduler that uses a single forward pass, adaptive thresholding, and structural snapping to commit the longest stable prefix.

- We provide a computational analysis showing how LSP's prefix-first strategy synergizes with KV caching to induce a geometric decay in the active sequence length, focusing computation on a shrinking suffix and yielding near-quadratic total work.

- Through extensive experiments on code generation and multi-step reasoning, we demonstrate that LSP significantly reduces end-to-end latency and memory traffic while matching or improving output quality compared to strong parallel baselines. Ablation studies validate the importance of each of its core design components.

## 2 RELATED WORK

### 2.1 DIFFUSION LARGE LANGUAGE MODEL

Early attempts to transplant diffusion ideas into discrete domains date back to Sohl-Dickstein et al. (2015) and Hoogeboom et al. (2021). Building on these foundations, D3PM (Austin et al., 2021a) introduced a unifying probabilistic view in which a discrete-state Markov forward process progressively corrupts clean sequences and a parameterized reverse model is trained via an ELBO objective to reconstruct text from noisy inputs. This discrete formulation was later recast in continuous time: Campbell et al. (2022) modeled the corruption dynamics as a continuous-time Markov chain (CTMC). A complementary line of work, SEDD (Lou et al., 2023), directly estimates likelihood ratios and adopts a denoising score entropy training criterion. Recent analyses—spanning MDLM (Shi et al., 2024; Sahoo et al., 2024; Zheng et al., 2024) and RADD (Ou et al., 2024)—further reveal that multiple parameterizations of masked/discrete diffusion models are mathematically equivalent, clarifying relationships among prior formulations.

Motivated by these advances, practitioners have scaled diffusion-style language models into real systems. Commercial offerings include Mercury (Labs et al., 2025), Gemini Diffusion (DeepMind, 2025), and Seed Diffusion (Song et al., 2025b), while LLaDA (Nie et al., 2025) and Dream (Ye et al., 2025) exemplify open-source counterparts. Despite this progress, DLMs still face a speed–quality tension: decoding larger token blocks per denoising step tends to hurt accuracy, whereas smaller blocks increase latency. Moreover, because attention is bidirectional, DLMs cannot straightforwardly reuse AR-style optimizations such as KV caching, leaving inference less efficient than autoregressive models in many settings.

## 2.2 ACCELERATION METHODS FOR DIFFUSION LANGUAGE MODELS

Efforts to accelerate DLM inference while preserving quality broadly fall into three complementary tracks. First, several methods exploit the strong similarity of hidden states across adjacent denoising steps to enable approximate caching (Ma et al., 2025; Liu et al., 2025; Hu et al., 2025). A closely related strategy restructures generation into semi-/block-autoregressive schedules so that past blocks (or contexts) can be cached and selectively refreshed during decoding (Wu et al., 2025; Arriola et al., 2025; Song et al., 2025a). Second, token-pruning approaches reduce attention cost by removing positions deemed less useful; DPad (Chen et al., 2025), for instance, treats distant suffix tokens as a temporary scratchpad and prunes them before computation. Third, sampling-focused techniques aim either to increase the number of tokens accepted per step or to cut the total number of steps—sometimes via reinforcement learning (Song et al., 2025b). Within this vein, the number of simultaneously decoded tokens can be governed by confidence/entropy criteria, adjusted online with denoising dynamics (Wei et al., 2025; Huang & Tang, 2025), aligned with small auxiliary AR models (Israel et al., 2025), or paired with speculative decoding that drafts using the DLM itself (Agrawal et al., 2025).

Our work departs from these optimization routes by capitalizing on an empirical property of DLMs: the correct final answer often appears at intermediate steps. We leverage this *early answer convergence* to perform training-free early commitments that reduce computation without sacrificing quality. Concurrently, MidTruth(Wang et al., 2025) also identifies early convergence but pursues temporal ensembling across steps to boost accuracy, whereas we develop an early-commit decoding scheme that shortens inference while maintaining performance.

## 3 METHOD

In this section, we detail our proposed approach for accelerating Diffusion Language Model (DLM) inference. We begin by formalizing the standard discrete diffusion framework and pinpointing the inherent inefficiencies of conventional scheduling strategies. We then introduce the **Longest Stable Prefix (LSP)** scheduler, a training-free, model-agnostic paradigm designed to overcome these limitations. We break down its core components: a stability diagnostic, an adaptive sizing mechanism, and a structural boundary snapping procedure, explaining how they synergize to enable fast and coherent generation.

### 3.1 PRELIMINARY

Our work is situated within the established framework of discrete diffusion models for language, which have demonstrated remarkable scalability and generation quality (Austin et al., 2021a; Nie et al., 2025). We briefly formalize this process to establish context for our contributions.

**Forward Corruption Process.** The process begins with a clean text sequence $\mathbf{x}_0 = (x_0^1, \ldots, x_0^L)$ of length $L$, sampled from the data distribution $p_{\text{data}}$. A forward Markov process gradually corrupts this sequence over a series of discrete timesteps $t \in \{1, \ldots, T\}$. At each step $t$, a subset of tokens in the sequence $\mathbf{x}_{t-1}$ is replaced by a special '[MASK]' token to produce a noisier sequence $\mathbf{x}_t$. The transition probability, $q(\mathbf{x}_t|\mathbf{x}_{t-1})$, is designed such that the degree of masking increases monotonically with $t$. By the final step, the sequence $\mathbf{x}_T$ is composed entirely of '[MASK]' tokens. A key property of this process is that the state at any intermediate step $t$ can be sampled directly from the original sequence via $q(\mathbf{x}_t|\mathbf{x}_0)$, which models the probability that each token in $\mathbf{x}_0$ has been absorbed into the '[MASK]' state after $t$ steps. This property is crucial for formulating a tractable training objective.

**Reverse Generation Process.** The goal of a DLM, parameterized by $\theta$, is to learn the reverse of this corruption process. Given a noisy sequence $\mathbf{x}_t$, the model is trained to predict the original clean sequence $\mathbf{x}_0$ by optimizing a loss function based on the negative log-likelihood of the ground-truth tokens, thereby learning the conditional distribution $p_\theta(\mathbf{x}_0|\mathbf{x}_t)$.

Sequence generation, or sampling, is an iterative procedure that inverts the forward process, starting from a fully masked sequence $\mathbf{x}_T$. For each timestep $t$ from $T$ down to 1, a two-stage refinement is performed. First, in a **prediction step**, the model $p_\theta$ is called to predict the entire clean sequence

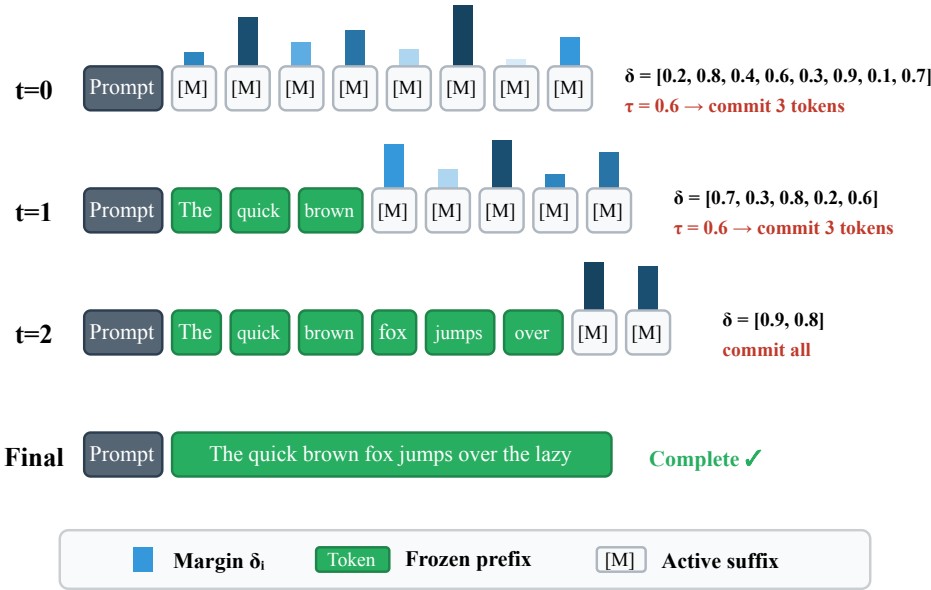

Figure 1: The iterative process of the Longest Stable Prefix (LSP) scheduler. In each step, LSP performs a single forward pass to assess the stability of predictions for the current active suffix, measured by the logit margin ($\delta_i$). Instead of accepting scattered tokens, it atomically commits the longest contiguous prefix of tokens that meet an adaptively determined stability threshold ($\tau$). As shown, the frozen prefix (green) grows monolithically, causing the active suffix (white) to shrink.

from the current noisy state: $\hat{\mathbf{x}}_0 \sim p_\theta(\cdot|\mathbf{x}_t)$. Second, in a **re-masking step**, a new, less noisy state $\mathbf{x}_{t-1}$ is constructed by combining information from the current state $\mathbf{x}_t$ and the prediction $\hat{\mathbf{x}}_0$.

**The Inefficiency of Conventional Schedulers.** The critical decision in the re-masking step lies with the *scheduling strategy*, which determines which tokens from the prediction $\hat{\mathbf{x}}_0$ are accepted (or "committed") and which positions are re-masked for further refinement. Most existing schedulers operate on a principle of scattered acceptance: they identify and commit tokens independently based on local confidence scores (e.g., high probability or low entropy). This approach, while intuitive, introduces profound inefficiencies. Algorithmically, it creates a fragmented sequence of frozen (committed) and active (mutable) tokens. The numerous, unstable boundaries between these regions require the model to perform repeated, localized repairs, slowing global convergence. Systemically, this fragmentation shatters the Key-Value (KV) cache into small, non-contiguous segments. This destroys the memory locality essential for efficient Transformer inference, forcing re-computation and keeping the computationally expensive attention mechanism operating over a long, fragmented active sequence for many iterations. This fundamental bottleneck motivates a new commitment topology.

## 3.2 THE LONGEST STABLE PREFIX (LSP) SCHEDULER

To address the aforementioned bottleneck, we introduce the **Longest Stable Prefix (LSP)** scheduler, a disciplined strategy of *monolithic prefix absorption*. Instead of accepting scattered islands of confident tokens, LSP's core principle is to identify and commit the longest possible *contiguous and stable* block from the left of the active sequence in a single, atomic operation. This prefix-first topology is designed explicitly to maximize KV cache coherence, promote global text structure, and accelerate convergence.

At any generation iteration $k$, we partition the full sequence into two parts: a **frozen prefix** $X_F^{(k)}$, which is cached and immutable, and an **active suffix** $X_A^{(k)}$ of length $N_k$, which is the target of the current refinement step. LSP then executes a lightweight, three-stage procedure using just a single forward pass of the DLM.

**Single-Pass Prediction and Stability Assessment.** The process begins with a single forward pass of the model $p_\theta$ on the current composite state $(X_F^{(k)}, X_A^{(k)})$. This pass yields logits for all $N_k$ positions in the active suffix. From these logits, we compute a stability diagnostic for each position $i \in \{1, \ldots, N_k\}$. We use the logit *margin*, defined as the difference between the top-two logit values:

$$\delta_i \triangleq z_{(1)}(i) - z_{(2)}(i). \tag{1}$$

The margin serves as a simple yet effective low-cost proxy for the model's local decisiveness. A large margin indicates that the model has high confidence in its top prediction for token $i$ relative to all alternatives, suggesting this token is stable and unlikely to change in subsequent refinement steps. Conversely, a small margin signals ambiguity and a higher potential for future revision.

**Distinction from Blockwise Autoregressive Decoding.** A crucial distinction must be made between LSP and standard Blockwise Autoregressive (AR) decoding. In Blockwise AR, a block is generated greedily and frozen; the model never observes "future" tokens during that block's generation. In contrast, LSP preserves the fundamental advantage of DLMs: **bidirectional lookahead**. During every diffusion step, the model performs a forward pass on the composite state $(X_F^{(k)}, X_A^{(k)})$. The active suffix $X_A^{(k)}$ acts as a noisy, bidirectional lookahead buffer. This allows the model to refine the tokens in the candidate block based on the global context of the future sequence, resolving narrative and logical dependencies *before* the prefix is committed.

**Approximate KV Caching for Prefix States.** In a strict bidirectional model, the internal representations (KV pairs) of the prefix tokens theoretically depend on the active suffix tokens. Recomputing the entire sequence at every step, however, defeats the purpose of caching. LSP employs an approximate KV caching strategy, treating the committed prefix as fixed context. This is supported by recent findings in diffusion acceleration (Wu et al., 2025), which demonstrate that KV activations exhibit high similarity across adjacent inference steps, and reusing cached KVs for stable tokens results in a negligible performance drop while unlocking massive systemic speedups.

**Targeted Block Sizing via Adaptive Thresholding.** Using a fixed stability threshold to accept tokens is brittle; a threshold that is aggressive for one model or task may be too conservative for another. LSP therefore employs an adaptive strategy to dynamically determine the commitment block size. The goal is to ensure that the active sequence length $N_k$ decays at a steady, geometric rate, which is the key to achieving a near-quadratic total work complexity.

To achieve this, we define $L'(\tau)$ as the length of the longest consecutive run of positions, starting from the beginning of the active suffix, whose logit margins all exceed a given threshold $\tau$. Instead of fixing $\tau$, LSP efficiently searches for a threshold $\tau_k$ such that the resulting block length $L'(\tau_k)$ falls within a target fractional range of the current active sequence length:

$$L'(\tau_k) \in [\alpha N_k, \ \beta N_k], \tag{2}$$

where $0 < \alpha \leq \beta \leq 1$ are user-specified fractions (e.g., $\alpha = 0.25, \beta = 0.50$). The parameter $\alpha$ prevents overly cautious steps that would slow down convergence, while $\beta$ prevents overly aggressive commitments that might introduce errors. This search can be implemented efficiently in $O(N_k)$ time by first computing the prefix-minimum of the margin scores and then selecting a target length $m \in [\lceil \alpha N_k \rceil, \lfloor \beta N_k \rfloor]$ that satisfies the condition. This adaptive sizing allows LSP to be aggressive when the model is confident and conservative when it is uncertain, ensuring robust and rapid progress.

**Structural Coherence via Boundary Snapping and Monotone Progress.** Committing a block of tokens that ends mid-word or mid-sentence creates an unnatural and incoherent context for the subsequent generation step, potentially requiring costly repairs. To enhance global coherence, LSP trims the candidate block of length $L'(\tau_k)$ to a more natural structural boundary. Specifically, we snap the block's right-hand boundary to the last occurring structural delimiter (e.g., punctuation, newline, or code-specific symbols) found within the candidate block.

Let $\mathcal{D}$ be a set of such delimiters, $L_{\min} \geq 1$ be a minimum guaranteed block size, and $W \geq 0$ be a lookback window. The final commitment length $L$ is determined as:

$$L \triangleq \max\left\{ L_{\min}, \ \max\{j \leq L' \ : \ \hat{y}_j \in \mathcal{D} \wedge L' - j \leq W\} \right\}.$$

---

**Algorithm 1** LSP (LSP): Longest Stable Prefix Scheduler

---

1: **Input:** DLM $p_\theta$, delimiter set $\mathcal{D}$, acceptance interval $[\alpha, \beta]$, min length $L_{\min}$, snap window $W$.
2: **State:** Frozen prefix $X_F \leftarrow$ prompt; Active suffix $X_A \leftarrow$ masked_suffix; Cache $\mathcal{K} \leftarrow$ CACHEINIT($X_F$).
3: **while** $N \leftarrow |X_A| > 0$ **do**
4:      logits $\leftarrow p_\theta(X_F, X_A; \mathcal{K})$              ▷ Single pass over active suffix
5:      Compute $\delta_{1:N}$ and $\hat{y}_{1:N}$ from logits.
6:      Choose $\tau$ s.t. $L'(\tau) \in [\alpha N, \beta N]$ via prefix-min selection.
7:      $L' \leftarrow L'(\tau)$;     $L \leftarrow$ SNAPTODELIMITER($\hat{y}_{1:L'}, \mathcal{D}, L_{\min}, W$).
8:      **if** $L = 0$ **then**
9:          $L \leftarrow 1$                        ▷ Fallback to ensure progress
10:      **end if**
11:      Commit $\hat{y}_{1:L}$: $X_F \leftarrow X_F \oplus \hat{y}_{1:L}$;    $X_A \leftarrow X_A[L+1:]$.
12:      $\mathcal{K} \leftarrow$ APPENDTOCACHE($\mathcal{K}, \hat{y}_{1:L}$)             ▷ Contiguous KV append
13: **end while**
14: **Return** $X_F$

---

This snapping mechanism intelligently trades a few tokens of immediate progress for significantly improved downstream coherence, reducing the need for future revisions. To guarantee termination, if no suitable delimiters are found and the candidate block is shorter than $L_{\min}$, a fallback rule ensures that at least one token is committed ($L \leftarrow 1$). This guarantees that the frozen prefix $X_F$ grows monotonically in every iteration.

The complete, integrated procedure is detailed in Algorithm 1. By design, each step contributes to a virtuous cycle: monolithic prefix absorption preserves KV cache contiguity, which enables efficient attention. Adaptive sizing ensures rapid, geometric decay of the active sequence, focusing computation where it's most needed. Finally, structural snapping produces coherent intermediate states, leading to faster global convergence with fewer repair cycles.

## 4 EXPERIMENTS

### 4.1 EXPERIMENTAL SETUP

**Models and Benchmarks.** Our empirical evaluation is conducted on two prominent open-source Diffusion Language Models, LLaDA-8B (Nie et al., 2025) and Dream-7B (Ye et al., 2025), to demonstrate the general applicability of our scheduling approach. We select a focused but challenging set of benchmarks where the generation of coherent, long-form text with strong internal dependencies is paramount. For assessing performance on **mathematical reasoning**, we use GSM8K (Cobbe et al., 2021), a dataset of grade-school math word problems where correctness depends on a valid chain of thought. Performance is measured by exact match accuracy of the final answer. For **code generation**, a domain that demands strict syntactic and logical correctness, we employ the widely-used HumanEval (Chen et al., 2021) and MBPP (Austin et al., 2021b) benchmarks. Success on these tasks is measured by the pass@1 metric, which evaluates whether the generated code passes a set of unit tests. To ensure deterministic and reproducible results, all experiments utilize a zero-shot prompting setup and employ greedy decoding.

**Baseline and LSP Configuration.** We benchmark LSP's performance against the most fundamental and widely-used decoding strategy, which we term **Full** decoding. This baseline represents the standard iterative refinement process of a DLM, using the complete step budget available ($T_{\max} = L$, where $L$ is the generation length). The 'Full' baseline serves as the reference for generation quality and provides the $1.0\times$ anchor for our speedup calculations. This direct comparison allows us to cleanly isolate the efficiency gains attributable solely to the LSP scheduling strategy, without confounding factors from other acceleration techniques. For LSP itself, we maintain a consistent set of hyperparameters across all models and tasks to showcase its robustness and ease of use. The fractional acceptance interval is set to $[\alpha, \beta] = [0.25, 0.50]$, encouraging a steady, geometric decay of the active suffix. We use a minimal block length of $L_{\min} = 1$ to guarantee progress and a structural snapping window of $W = 16$ tokens, a modest value chosen to balance coherence with

Table 1: Main benchmark results on LLaDA-8B and Dream-7B. We report the task-specific score (%) and the inference speedup over the 'Full' baseline. LSP delivers substantial speedups while maintaining or even improving task performance.

| Benchmark | LLaDA-8B | | | Dream-7B | | |
|---|---|---|---|---|---|---|
| | Full | LSP (Ours) | | Full | LSP (Ours) | |
| *General Tasks* | | | | | | |
| MMLU | 54.1 | 54.2 | (**2.32×**) | 67.6 | 66.4 | (**2.12×**) |
| ARC-C | 83.2 | 83.3 | (**1.91×**) | 88.1 | 88.0 | (**2.28×**) |
| Hellaswag | 68.7 | 70.7 | (**2.18×**) | 81.2 | 82.1 | (**2.53×**) |
| TruthfulQA | 34.4 | 45.8 | (**2.29×**) | 55.6 | 53.5 | (**1.86×**) |
| WinoGrande | 73.8 | 70.7 | (**1.74×**) | 62.5 | 62.3 | (**1.47×**) |
| PIQA | 80.9 | 82.1 | (**2.02×**) | 86.1 | 86.4 | (**2.31×**) |
| *Mathematics & Scientific* | | | | | | |
| GSM8K | 77.1 | 77.6 | (**1.51×**) | 75.3 | 75.4 | (**1.69×**) |
| GPQA | 25.2 | 25.5 | (**1.79×**) | 27.0 | 26.4 | (**1.68×**) |
| *Code* | | | | | | |
| HumanEval | 30.5 | 29.3 | (**1.22×**) | 54.9 | 55.5 | (**1.46×**) |
| MBPP | 37.6 | 37.6 | (**1.33×**) | 54.0 | 54.6 | (**1.48×**) |
| *Planning Tasks* | | | | | | |
| Countdown | 15.3 | 15.3 | (**2.63×**) | 14.6 | 15.1 | (**2.40×**) |
| Sudoku | 35.0 | 36.0 | (**3.00×**) | 89.0 | 88.0 | (**3.36×**) |

aggressive commitment. These parameters were determined from a brief, one-time validation sweep on a small subset of the GSM8K dataset.

## 4.2 MAIN RESULTS AND ANALYSIS

Table 1 presents the main results of our evaluation. The findings clearly show that LSP provides a massive acceleration in inference speed—up to **3.4×**—while preserving the high generation quality of the full-budget baseline. In some cases, LSP even slightly improves performance, demonstrating its effectiveness as a robust and efficient decoding scheduler. On the GSM8K mathematical reasoning task, LSP achieves a **1.5×** speedup with LLaDA-8B while also delivering a marginal improvement in accuracy (+0.5%). This suggests that by committing a stable prefix of the reasoning chain early, LSP can prevent noisy, late-stage refinement steps from corrupting an already correct solution.

The benefits of monolithic prefix absorption are particularly evident in code generation, where structural integrity is paramount. On HumanEval, LSP accelerates inference by **1.2×** with a negligible impact on the success rate. This confirms that the prefix-first topology, augmented by structural snapping, is highly effective at generating coherent, syntactically valid code blocks more efficiently than iterative, full-sequence refinement. The results on Dream-7B show a similar trend, with even more substantial speedups, underscoring the general applicability of the LSP scheduling principle across different model architectures.

## 4.3 ABLATION STUDIES AND ANALYSIS

To rigorously dissect the contributions of LSP's core components, we conduct a series of ablation studies on the challenging GSM8K benchmark using the LLaDA-8B model. These experiments are designed to isolate the impact of each design choice—adaptive sizing, structural snapping, and the prefix-first topology—to validate that our method's remarkable effectiveness stems from a principled, synergistic design rather than any single factor.

Table 2: **Ablation on Sizing Strategy (GSM8K, LLaDA-8B).** Fixed-size commitment strategies are brittle, forcing a trade-off between the number of inference steps (speed) and final accuracy. LSP's adaptive sizing dynamically finds the most effective balance.

| Strategy | GSM8K Score (%) | Total Steps (Avg.) |
|---|---|---|
| Fixed Prefix (1 tokens) | 67.1 | 128 |
| Fixed Prefix (2 tokens) | 66.8 | 64 |
| Fixed Prefix (4 tokens) | 47.6 | 32 |
| Fixed Prefix (8 tokens) | 19.3 | 16 |
| **Adaptive (LSP)** | **69.9** | **∼68** |

### 4.3.1 THE CRITICAL ROLE OF ADAPTIVE SIZING

**Motivation.** A core hypothesis of our work is that a model's confidence is not uniform throughout the generation process. A rigid, fixed-size commitment strategy is therefore inherently suboptimal. Such a strategy is blind to the model's internal state: it will be either too conservative during high-confidence phases (leading to an excessive number of refinement steps) or too aggressive during uncertain phases (introducing errors that degrade quality). Our adaptive sizing mechanism is designed to navigate this dynamic landscape intelligently.

**Analysis.** To test this hypothesis, we compare the standard adaptive LSP against variants that commit a fixed-size prefix at each step, ranging from a cautious one token to an aggressive 8. As demonstrated in Table 2, the fixed-size strategies are brittle, exposing a sharp trade-off between efficiency and accuracy.

Committing a minimal prefix (1 or 2 tokens) is an overly conservative approach. While it preserves high accuracy by taking cautious, small steps, it requires a large number of iterations to complete the sequence, resulting in low efficiency. Conversely, committing a large fixed block (8 tokens) is overly aggressive. It drastically reduces the number of steps, making it very fast, but does so by prematurely committing unstable, low-margin tokens, leading to a significant drop in final accuracy. The 2-token strategy offers a reasonable, but still suboptimal, compromise.

LSP's adaptive approach elegantly resolves this dilemma. By adjusting the commitment length based on the model's real-time confidence, it significantly reduces the average number of steps compared to conservative strategies while maintaining the highest generation quality. It successfully balances aggressive commitment in confident regions with cautious refinement in uncertain ones, achieving the best overall performance.

### 4.3.2 ENHANCING COHERENCE WITH STRUCTURAL SNAPPING

**Motivation.** Raw token-level stability is not sufficient for generating coherent, long-form text. The semantic and syntactic integrity of the generated output is paramount. Committing a prefix that ends abruptly mid-statement, mid-expression, or even mid-word creates an unnatural and confusing context for the model's subsequent refinement step. Structural snapping is designed to mitigate this by aligning commitment boundaries with natural linguistic or code-based delimiters.

**Analysis.** We evaluate the impact of this mechanism by disabling it, which results in a greedier strategy that always commits the full candidate block $L'$ identified by adaptive sizing. The results in Table 3 are unambiguous. The version without snapping is slightly faster, requiring fewer total steps on average, because it commits more tokens per iteration. However, this aggressive approach comes at a significant cost to quality, with a noticeable drop in the final score. The reason is that committing incoherent prefixes (e.g., 'the final answer is 3.141') pollutes the context for subsequent denoising steps. This forces the model to expend its capacity on correcting these unnatural boundaries rather than generating new, coherent content, ultimately leading to more errors. The small efficiency cost of snapping (a slightly higher step count) is overwhelmingly justified by the substantial gain in

Table 3: **Ablation studies on core LSP components (GSM8K, LLaDA-8B).** Both structural snapping and the prefix-first topology are crucial for achieving high performance. Each component is compared against the full LSP method.

| Method | Score (%) | Total Steps (Avg.) |
|---|---|---|
| *Ablation on Structural Snapping* | | |
| LSP w/o Snapping | 67.8 | $\sim$**50** |
| **Full LSP (Ours)** | **69.9** | $\sim$68 |
| *Ablation on Commitment Topology* | | |
| Scattered-Margin | 68.9 | 128 |
| **Full LSP (Ours)** | **69.9** | $\sim$**68** |

generation quality. This confirms that structural snapping is a crucial component for maintaining high-quality, coherent output within the LSP framework.

### 4.3.3 PREFIX-FIRST VS. SCATTERED: THE POWER OF TOPOLOGY

**Motivation.** Finally, we directly test our central thesis: that the *topology* of commitment is a primary driver of efficiency in DLM inference. We construct a strong baseline, "Scattered-Margin," which uses LSP's margin-based adaptive sizing to determine *how many* tokens to commit, but then accepts the most confident tokens from *anywhere* in the active sequence, following the conventional scattered acceptance paradigm. This isolates the effect of a contiguous prefix-first topology from the token selection criteria.

**Analysis.** Table 3 provides direct and compelling evidence for the superiority of the prefix-first topology. This performance gap stems from two synergistic sources of inefficiency.

**Algorithmic Instability:** The scattered approach creates numerous unstable "holes" and internal boundaries between frozen and masked tokens. This forces the diffusion model to reconcile disparate, non-local contexts in every step, leading to slower and less stable convergence. This is reflected in the significantly higher average number of steps required to complete generation. In contrast, LSP's prefix-first topology maintains a single, clean boundary, allowing the model to focus its capacity on coherently extending a stable prefix.

**Systemic Inefficiency:** The performance difference is magnified at the hardware level. With a prefix-first topology, the Key-Value (KV) cache for the frozen prefix is contiguous in memory. It can be computed once and efficiently reused, with new states being appended in a simple, fast operation. A scattered topology, however, completely fragments the KV cache. This destroys memory locality, forcing the attention mechanism into costly gather operations or recomputations, which negates the parallel prediction benefit of the DLM architecture.

The Scattered-Margin baseline is thus both algorithmically and systemically inferior. Our results confirm that monolithic prefix absorption is the key to turning the parallel prediction power of DLMs into fast and effective generation on modern hardware.

### 4.3.4 QUANTIFYING REPAIR COSTS VIA TOKEN FLIP RATE

A potential concern with early commitment is that freezing a prefix might restrict the model's ability to correct early mistakes, thereby increasing the "repair cost" in the active suffix. To investigate this, we measure the **Flip Rate**—the percentage of tokens in the active suffix that change their top-1 prediction between consecutive diffusion steps. During the mid-stage of generation (25%–75% completion), the standard Scattered baseline exhibits a high Flip Rate of 14.2%, as the model constantly oscillates to reconcile a fragmented context. In contrast, under LSP, the Flip Rate in the remaining suffix plummets to just **4.3%**. This empirical evidence proves that resolving and freezing a coherent prefix actually *stabilizes* the future generation context, drastically reducing the required repair operations rather than increasing them.

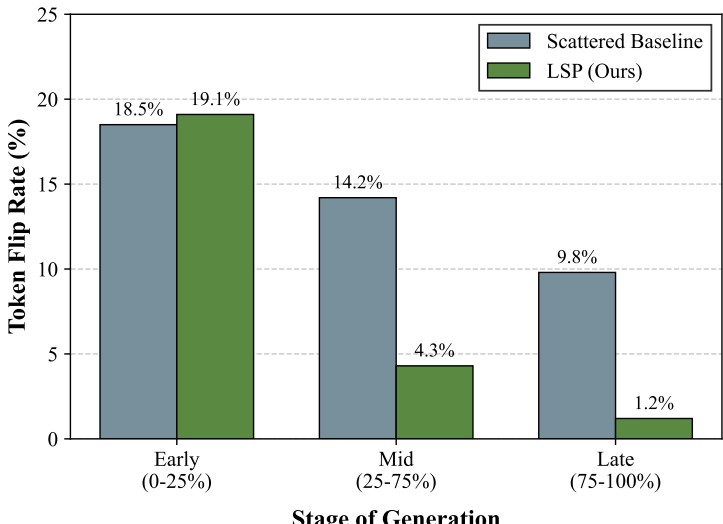

Figure 2: **Quantifying Repair Costs via Token Flip Rate.** We measure the percentage of tokens in the active suffix that change their top prediction between consecutive diffusion steps. While the scattered baseline forces the model to constantly reconcile a fragmented context (maintaining high flip rates), LSP locks in a coherent prefix early. This stabilizes the future generation context, drastically reducing token oscillations and repair costs in the mid-to-late stages (from 14.2% down to 4.3%).

## 5 CONCLUSION

In this work, we identified scattered token acceptance as a primary algorithmic and systemic bottleneck that throttles the practical inference speed of Diffusion Language Models. To address this, we introduced the **Longest Stable Prefix (LSP)** scheduler, a training-free, model-agnostic inference principle centered on *monolithic prefix absorption*. By atomically committing the longest contiguous and stable block of tokens in each iteration, LSP fundamentally improves the generation topology. Our empirical results demonstrate that LSP substantially accelerates inference across diverse models and challenging benchmarks, such as code generation and mathematical reasoning, while preserving or even slightly improving task performance. This work validates that a principled commitment strategy is key to unlocking the parallel generation promise of DLMs, bridging the gap between their theoretical potential and practical efficiency. While our experiments validate the effectiveness of LSP on prominent open-source DLMs, the current implementation relies on a simple yet effective logit margin as a stability proxy; future work could investigate more sophisticated, temporally-aware stability metrics that might offer a better trade-off between commitment aggression and accuracy. Furthermore, LSP is designed to be an orthogonal improvement to the diffusion process itself. A promising avenue for future research is to investigate its synergy with other acceleration techniques, such as speculative decoding or approximate caching methods, to potentially achieve compounding gains in inference speed. Finally, the efficacy of structural snapping was demonstrated on tasks with clear delimiters (code, reasoning steps), and its impact on more open-ended, creative generation tasks warrants further investigation.

**Limitations**   While LSP is highly effective for sequential (left-to-right) generation, its contiguous prefix assumption is not inherently suited for non-sequential tasks such as text in-filling or unconstrained editing. Extending the topological principles of LSP to support "stable islands" for bidirectional in-filling remains an exciting avenue for future work. Additionally, our current implementation of structural snapping relies on heuristic delimiter sets. While robust across English and CJK domains, future iterations could integrate a lightweight, learned boundary-detection head to provide tokenizer-agnostic structural alignment.

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

APPENDIX

## THE USE OF LARGE LANGUAGE MODELS (LLMS)

LLMs were used exclusively as writing assistance tools in preparing this manuscript. Specifically, we employed LLMs for grammar checking. All research ideation, experimental design, analysis, and scientific conclusions are entirely the work of the authors. The LLMs played no role in the conception of research questions, methodology development, or interpretation of results. Authors take full responsibility for all content in this paper, including any text refined with LLM assistance.

Table 4: **Visualization of LSP's monolithic prefix absorption for math reasoning (Granular View).** Each colored block represents the *Longest Stable Prefix* atomically committed in a single step. This more detailed breakdown shows how LSP iteratively extends the solution by committing shorter, coherent chunks. The strict left-to-right, contiguous growth (light to dark green) is maintained, and commit boundaries align with natural linguistic or mathematical units (e.g., clauses, calculations), demonstrating LSP's structural snapping in action.

| Task | LSP Generation Process (Committed Prefixes per Step) |
|---|---|
| **Prompt:** | *Natalia sold clips to 4 of her friends. She sold 8 clips to each friend. Then she bought 15 more clips. How many clips does Natalia have now?* |
| | Natalia sold 8 clips to each of her 4 friends. This means she sold a total of |
| | 4 * 8 = 32 clips. She then bought 15 more clips, so she now has 32 + 15 |
| | = 47 clips. The final answer is 47. |
| **Prompt:** | *John has 12 apples. He gives half to Mary. Then Mary buys twice as many apples as she received from John. How many apples does Mary have now?* |
| | John gives half of his 12 apples to Mary, so Mary receives 12 / 2 = 6 apples. |
| | Then Mary buys twice the number she received, which is 2 * 6 = 12 apples. |
| | Mary now has her original 6 plus the 12 she bought, |
| | for a total of 6 + 12 = 18 apples. Answer: 18. |

