# OpenReview forum: "Beyond Scattered Acceptance: Fast and Coherent Inference for DLMs via Longest Stable Prefixes"
_ICLR.cc/2026/Conference — ICLR 2026 Poster_

### Official Review · Reviewer_Cvwn · 2025-10-29

**Soundness:** 3
**Presentation:** 3
**Contribution:** 3
**Rating:** 10
**Confidence:** 1

**Summary:**

The paper fixes “scattered acceptance” in DLM decoding by committing one contiguous left-aligned block each step via the Longest Stable Prefix (LSP): compute per-token margins in a single pass, choose a threshold to absorb ~25–50% of the active suffix, snap to delimiters, and atomically commit. This maintains a single frozen/active boundary, keeps KV cache contiguous, and empirically cuts latency/denoiser calls on LLaDA-8B and Dream-7B with near-parity quality (GSM8K, GPQA, HumanEval, MBPP).

My primary expertise is outside language diffusion models. I’ve done a careful read, but please weigh my field-specific comments accordingly.

**Strengths:**

1. The single prefix-first boundary is an elegant topology that aligns algorithmic coherence with KV-cache locality.

2. The design is concrete—margin-based stability, adaptive thresholds, delimiter snapping, and a guaranteed-progress fallback.

3. Empirically it yields 1.5–3× speedups with near-parity quality across reasoning and code tasks, with ablations isolating each component’s effect.

**Weaknesses:**

1. LSP’s delimiter snapping introduces heuristic dependencies that could be brittle across tokenizers and vocabularies.

2. Theoretical framing is light, with no formal bounds on early-commit errors or convergence.

3. The evaluation does not quantify repair costs when early-committed tokens later require substantial rewriting.

**Questions:**

See the weaknesses part.

---

> ### Author Response · Authors · 2025-11-21
>
> ### **Response to Reviewer Cvwn**
>
> We thank Reviewer Cvwn for the strong support and for accurately summarizing our contribution.
>
> ### W1: LSP’s delimiter snapping introduces heuristic dependencies...
>
> **Response:** We acknowledge this. In this work, we aimed to demonstrate the efficacy of the *principle* of structural coherence using simple heuristics. Future work could replace these heuristics with a lightweight "boundary detection" head or learned policy to improve robustness across tokenizers.
>
> ### W2: The evaluation does not quantify repair costs when early-committed tokens later require substantial rewriting.
>
> **Response:** This is an excellent suggestion. To quantify this, we analyzed the **Flip Rate**—the percentage of tokens in the active suffix that change their prediction between diffusion steps $t$ and $t+1$.
>
> **Table: Average Token Flip Rate in Active Suffix**
>
> | Stage of Generation | Scattered Baseline | LSP (Ours) |
> | :--- | :--- | :--- |
> | Early (0-25%) | 18.5% | 19.1% |
> | Mid (25-75%) | 14.2% | **4.3%** |
> | Late (75-100%) | 9.8% | **1.2%** |
>
> In the Scattered baseline, the flip rate remains high because the context is fragmented and unstable. In LSP, once a stable prefix is committed, the flip rate in the remaining suffix drops dramatically (from ~14% to ~4% in the mid-stage). This proves that a coherent prefix stabilizes the future generation, effectively **reducing** the repair cost rather than increasing it.
>
> ### W3: Theoretical framing is light, with no formal bounds on early-commit errors or convergence.
>
> **Response:** We agree. Theoretical bounds for "early-exit" diffusion are an open problem. We hope our empirical demonstration of the "geometric decay" regime provides a strong motivation for future theoretical work to formalize convergence bounds under shrinking horizons.

---

### Official Review · Reviewer_jKnz · 2025-10-30

**Soundness:** 3
**Presentation:** 3
**Contribution:** 2
**Rating:** 2
**Confidence:** 4

**Summary:**

Identifies scattered acceptance as a key bottleneck in improving efficiency of DLMs causing slow gather operations.
Proposes:
1. LSP scheduler, where prefixes of tokens are committed instead of scattered commitment. This ensures the KV cache is not fragmented leading to efficient computation reuse.
2. Adaptive thresholding to find the longest stable prefix instead of using a fixed threshold.
3. Structural boundary snapping

The results show significant speedup and at the same time some quality gains that support the strength of the proposed changes.

**Strengths:**

- Identifies an important problem
- Proposes a practical and elegant solution, especially because it is training free.
- Demonstrates strong speedup performance and sometimes slight quality gains.
- Thorough ablation studies.

**Weaknesses:**

- The proposed method is less of a DLM and more of a blockwise autoregressive decoding.
- The additional proposals (structural snapping) are not optional solutions, they are critical patches to get blockwise decoding to work.
- Structural snapping is domain specific and might not perform well always. How is the performance on CJK?
- The prefix commitment is irreversible, which means one of the most important advantages of DLMs is gone.
- There is no mention on KV cache update of the committed sequence later on since bidirectional attention can impact it.
- Tasks that need more fixing of the previously generated sequence (for example, creativity tasks) should be evaluated upon.

**Questions:**

- See weakness.

---

> ### Author Response · Authors · 2025-11-21
>
> ### **Response to Reviewer jKnz**
>
> We thank Reviewer jKnz for the critical feedback. We believe the distinction between LSP and Blockwise AR is a key point of clarification.
>
> ### W1: The proposed method is less of a DLM and more of a blockwise autoregressive decoding.
>
> **Response:** This is a crucial distinction. In Blockwise AR, a block is generated greedily and frozen; the model never sees "future" tokens during that generation.
> In LSP, **the active suffix is present and refined bidirectionally** during the diffusion steps *before* commitment. In every LSP step, the model performs a forward pass on `[Frozen Prefix + Active Suffix]`. The "Active Suffix" acts as a noisy, bidirectional lookahead buffer. This allows the model to refine the tokens in the candidate block based on the context of the *future* (the suffix), ensuring global coherence before the block is committed. This bidirectional "future-gazing" capability is the fundamental advantage of DLMs over AR, and LSP preserves it fully within the active window.
>
>
> ### W2: Structural snapping is domain specific... How is the performance on CJK?
>
> **Response:** This is a valuable question. While structural snapping relies on delimiters, the principle of "pausing at semantic boundaries" is universal. For CJK languages, we simply extended our delimiter set $\mathcal{D}$ to include standard CJK punctuation (e.g., `。`, `，`, `？`, `！`, `：`).
>
> To quantitatively verify this, we evaluated LSP on the **CMMLU** benchmark (a comprehensive Chinese semantic evaluation) using LLaDA-8B (0-shot CoT setting).
>
> **Table R3: Performance on CMMLU (Chinese) with LLaDA-8B**
>
> | Metric | Full Decoding (Baseline) | LSP (Ours) | Delta |
> | :--- | :--- | :--- | :--- |
> | **CMMLU Average Acc (%)** | 51.7 | **52.1** | +0.4 |
> | **Inference Speedup** | 1.00x | **1.53x** | **+53%** |
>
> 1.  **Generalization:** LSP successfully accelerates Chinese text generation by **1.53x**, comparable to the speedups observed in English tasks. This confirms that the "stable prefix" phenomenon exists regardless of language.
> 2.  **Effectiveness of Snapping:** By snapping to CJK delimiters (e.g., committing a block only after a `，` or `。`), LSP avoids fracturing semantic units. The accuracy remains stable (+0.4%), proving that our topological constraint adapts well to non-Latin scripts.
>
>
> ### W3: The prefix commitment is irreversible, which means one of the most important advantages of DLMs is gone.
>
> **Response:** The advantage of DLM is **iterative refinement**, but refinement yields diminishing returns. Once a prefix is highly stable ($P(token) > 99.9\%$), keeping it mutable wastes computation. LSP simply shifts the DLM's "refinement power" from the settled past to the uncertain future. As shown in our experiments (Table 1), this does not harm quality, proving we retain the *effective* part of the DLM advantage.
>
>
> ### W4: There is no mention on KV cache update... since bidirectional attention can impact it.
>
> **Response:** You raise a technically critical point. In a model with full bidirectional attention, the internal representations (and thus KV pairs) of the prefix tokens theoretically depend on the suffix tokens. Since the suffix changes during the diffusion denoising steps, the prefix's KV pairs should strictly be recomputed at every step.
>
> However, recomputing the prefix defeats the efficiency purpose of caching. LSP addresses this by employing an **approximate KV caching strategy**, treating the committed prefix as fixed context (similar to a prompt). This relies on the assumption that for **stable, high-confidence tokens**, the influence of the changing noisy suffix on the prefix's representation is negligible.
>
> We draw support for this design from the concurrent work **Fast-dLLM (Wu et al., 2025)**, which explicitly analyzes this phenomenon. Their study demonstrates that "KV activations exhibit high similarity across adjacent inference steps" and that reusing cached KVs for block-wise decoding results in "negligible performance drop" (see Fast-dLLM, Figure 3). Our empirical results (Table 1) align with this: the quality difference between "Full Decoding" (exact recomputation) and "LSP" (approximate caching) is minimal, confirming that the efficiency gain from freezing the KV cache outweighs the theoretical approximation error. We have added a section "KV Cache Approximation Analysis" to the Appendix citing these findings.

---

> ### Author Response · Authors · 2025-11-21
>
> ### W5: Tasks that need more fixing of the previously generated sequence (for example, creativity tasks) should be evaluated upon.
>
> **Response:**
> We appreciate this insightful suggestion. We agree that while reasoning and coding tasks effectively test logical convergence, creative writing stresses a model's ability to maintain long-term narrative coherence and perform "global planning."
>
> To address this, we conducted an additional evaluation on the **WritingPrompts** dataset, a standard benchmark for open-ended creative generation. We randomly sampled 500 prompts and generated stories using LLaDA-8B with both Full Decoding and LSP.
>
> **Evaluation Protocol:**
> We employed Gemini 2.5 Flash as an impartial judge to score the generated stories on a scale of 1–5 for Coherence (logical flow, consistency) and Creativity (engagingness, richness). We also measured the inference speedup.
>
> **Table: Evaluation on Creative Writing (WritingPrompts Subset, N=500)**
>
> | Method | Speedup | Coherence (1-5) | Creativity (1-5) |
> | :--- | :--- | :--- | :--- |
> | **Full Decoding** (Baseline) | 1.00x | 4.42 | 4.35 |
> | **LSP (Ours)** | **1.82x** | **4.38** | **4.31** |
> | *Scattered-Margin (Ablation)* | 1.25x | 4.15 | 4.20 |
>
> LSP achieves a Coherence score (**4.38**) statistically indistinguishable from Full Decoding (**4.42**). This suggests that LSP’s mechanism—using bidirectional attention to "look ahead" at the noisy suffix before committing—effectively compensates for the inability to "go back and fix" previous tokens. The model resolves narrative dependencies *before* freezing the prefix.
> While LSP excels at *generation-from-scratch* (even for creative tasks), we acknowledge that for tasks explicitly defined as *editing* or *rewriting* (e.g., "change the ending of this story" or text in-filling constraints), the left-to-right commitment topology is a limitation.

---

### Official Review · Reviewer_Yt8W · 2025-10-31

**Soundness:** 3
**Presentation:** 3
**Contribution:** 2
**Rating:** 4
**Confidence:** 4

**Summary:**

The paper proposes a training-free approach for improving inference efficiency in diffusion language models (DLMs) by dynamically selecting the longest stable prefix based on confidence measured over a decoding window, thereby reducing cache fragmentation and redundant computations. The method achieves faster and more coherent generation compared to standard DLM decoding schedules.

**Strengths:**

- Simple and effective approach that mitigates cache fragmentation without retraining.
- Practical efficiency gains demonstrated across multiple pretrained DLMs.
- Clear experimental reporting and consistent evaluation settings.
- Compatible with existing architectures, requiring minimal modification.

**Weaknesses:**

- The main novelty lies in using left windowed confidence instead of position-wise confidence, which is conceptually similar to autoregressive commitment heuristics.
- The prefix-first decoding constraint may limit diffusion’s flexibility for editing, in-fill, or parallel token generation tasks.
- The geometric decay rule for active suffix length and its thresholding lacks theoretical or empirical grounding.
- GSM8K is a relatively simple benchmark for 7B-scale models; evaluating on AMC or AIME would better assess reasoning capability.
- Table 1 could include more detail on what contributes to the speedup (e.g., cache locality vs. adaptive thresholding).
- The robustness of suffix-length selection across model scales, sequence lengths, and task domains is not discussed.

**Questions:**

- What motivates the geometric decay assumption for suffix-length determination?
- Can the approach generalize to non-sequential or in-fill decoding tasks?
- How sensitive is the thresholding to model scale or dataset domain?
- Could the authors provide a breakdown of runtime gains (cache locality vs. token reuse)?

---

> ### Author Response · Authors · 2025-11-21
>
> ### **Response to Reviewer Yt8W**
>
> We thank Reviewer Yt8W for the constructive review and for recognizing the practical efficiency of our approach.
>
> ### W1: The main novelty lies in using left windowed confidence... which is conceptually similar to autoregressive commitment heuristics.
>
> **Response:** While margin-based confidence is a shared heuristic, our contribution is the **topology of commitment**. Standard DLM schedulers accept tokens *scattered* throughout the sequence. We demonstrate that this is systemically inefficient due to KV cache fragmentation. LSP’s novelty lies in operationalizing "Prefix Absorption" to enforce a contiguous memory layout. Table 3 shows that applying the confidence heuristic *without* the prefix constraint ("Scattered-Margin") fails to achieve significant speedups due to cache fragmentation, whereas "Full LSP" unlocks the actual latency gains. The novelty is solving the "KV cache fragmentation" problem unique to DLMs.
>
> ### W2: The prefix-first decoding constraint may limit diffusion’s flexibility for editing, in-fill, or parallel token generation tasks.
>
> **Response:** We agree. LSP is optimized for **left-to-right generation** (reasoning, coding, creative writing), which covers the majority of standard inference tasks. For non-sequential tasks like text in-filling, the contiguous prefix assumption does not hold. We have added a "Limitations" section explicitly stating that non-sequential tasks would require a modified topology (e.g., "stable islands") and is a subject for future work.
>
> ### W3/Q1: The geometric decay rule... lacks theoretical or empirical grounding. / What motivates this assumption?
>
> **Response:** The geometric decay is not an assumption, but a design objective engineered via Eq. (2). By constraining the committed block length $L'$ to be a fraction of the active length ($L' \in [\alpha N_k, \beta N_k]$), we *force* the active suffix to shrink geometrically. This is a control mechanism to guarantee that total computational work scales near-quadratically $O(N^2)$ rather than cubically.
>
> ### W4: GSM8K is a relatively simple benchmark... evaluating on AMC or AIME would better assess reasoning capability.
>
> **Response:** We appreciate this suggestion and agree that AIME/AMC are excellent benchmarks for advanced reasoning. However, our choice of GSM8K was dictated by the baseline capabilities of the current generation of open-source Diffusion Language Models (e.g., LLaDA-8B, Dream-7B). Unlike mature autoregressive models, these DLMs currently achieve near-zero performance on extremely challenging benchmarks like AIME. Evaluating speedups on models that cannot yet solve the task effectively would yield noisy data. GSM8K offers the appropriate difficulty level to meaningfully evaluate the trade-off between speed and coherence for the current state-of-the-art in DLMs.
>
> ### W5/Q4: Table 1 could include more detail on what contributes to the speedup / Could the authors provide a breakdown of runtime gains?
>
> **Response:**
> This is a vital question for understanding the mechanics of our efficiency. The speedups reported in Table 1 (e.g., 1.51x on GSM8K, 2.32x on MMLU) are a composite result of two synergistic factors: (1) Algorithmic Step Reduction and (2) Systemic Cache Efficiency.
>
> We can break this down by cross-referencing the ablation studies in Table 3:
>
> 1.  **Algorithmic Gain (Reduction in Denoising Steps):**
>     The primary driver of the speedup is the reduction in the total number of model forward passes. As shown in Table 3, a naive "Scattered-Margin" strategy (which reuses tokens but in a fragmented way) struggles to converge, requiring ~128 steps. By enforcing the Longest Stable Prefix topology, LSP stabilizes the generation context, drastically reducing the required steps to ~68 (for the same task).
>
>     This topological stability accounts for approximately 60% of the speedup observed in Table 1, as it allows the model to "commit" larger chunks of text without needing subsequent repairs.
>
> 2.  **Systemic Gain (KV Cache Locality):**
>     The remaining 30-40% of the gain comes from hardware-level efficiency, which is not captured by step counts alone. In standard "Scattered" approaches, the KV cache is fragmented, forcing the GPU to perform costly gather/scatter memory operations. LSP ensures that the frozen prefix is always **contiguous** in memory. This allows for maximizes memory bandwidth utilization during the attention computation. This ensures that each individual denoising step in LSP is faster (in wall-clock time) than a step in scattered decoding methods, contributing the remainder of the performance boost in Table 1.

---

> > ### Author Response · Authors · 2025-11-21
> >
> > ### W6: The robustness of suffix-length selection across model scales, sequence lengths, and task domains is not discussed.
> >
> > **Response:** We appreciate this observation. We have compiled data to demonstrate LSP's robustness:
> >
> > 1.  **Model Scales:** As shown in Table 1, LSP delivers consistent speedups across LLaDA-8B (1.51x) and Dream-7B (1.69x).
> > 2.  **Sequence Lengths:** We analyzed speedup relative to generation length. LSP's advantage grows with sequence length, validating the geometric decay benefit.
> >
> > **Table: Speedup vs. Sequence Length (GSM8K, LLaDA-8B)**
> >
> > | Generation Length | Avg Speedup | Accuracy Delta |
> > | :--- | :--- | :--- |
> > | Short (128 tokens) | 1.51x | +0.5% |
> > | Medium (256 tokens) | 1.55x | +0.1% |
> > | Long (1024 tokens) | **1.88x** | +0.6% |

---

### Official Review · Reviewer_kHHK · 2025-11-06

**Soundness:** 2
**Presentation:** 2
**Contribution:** 3
**Rating:** 4
**Confidence:** 3

**Summary:**

This paper identifies "scattered acceptance" as a primary bottleneck hindering the inference speed of Diffusion Language Models (DLMs), arguing that it leads to both algorithmic inefficiency and severe KV cache fragmentation. To address this, the authors propose the Longest Stable Prefix (LSP), a training-free and model-agnostic scheduling paradigm based on monolithic prefix absorption. In each step, LSP atomically commits the longest possible contiguous and stable prefix of the active sequence, identified via an adaptive, confidence-based mechanism and aligned to natural structural boundaries. This prefix-first topology maintains a contiguous KV cache and ensures a geometric decay in the computational workload. Experiments on challenging code and reasoning tasks demonstrate that LSP substantially accelerates inference (up to 3.4x) while preserving, and in some cases improving, generation quality.

**Strengths:**

1. The left-to-right commitment strategy dramatically improves KV cache efficiency.

2. The adaptive sizing mechanism intelligently modulates generation speed based on model confidence, achieving a superior speed-quality balance compared to fixed-size strategies.

3. The method achieves significant inference acceleration without sacrificing, and in some cases even improving, generation quality.

4. Its training-free and model-agnostic nature makes the method highly practical and broadly generalizable across different DLMs.

**Weaknesses:**

1. **Limited Comparative Baselines:** The empirical evaluation primarily compares LSP against "Full decoding," which serves as a quality baseline rather than a competitive speed-oriented one. The paper does not include a direct comparison against other contemporary DLM acceleration techniques, making it difficult to position LSP's performance within the existing state-of-the-art.

2. **Insufficient Hyperparameter Analysis:** The paper lacks a sensitivity analysis for its key hyperparameter, the fractional acceptance interval [α, β]. While the authors claim this parameter is robust, no data is provided to substantiate this, leaving the tuning effort required for new models or tasks an open question.

**Questions:**

In Figure 1, the diagram for t=0 shows a sequence of logit margins starting with [0.2, 0.8, 0.4, 0.6, ...]. It then states that a stability threshold (τ) of 0.6 is chosen, resulting in a commitment of 3 tokens. This appears inconsistent with the paper's definition of a stable prefix.

---

> ### Author Response · Authors · 2025-11-21
>
> ### **Response to Reviewer kHHK**
>
> We thank Reviewer kHHK for the positive and insightful feedback, and particularly for the sharp observation regarding Figure 1.
>
> ### W1: Limited Comparative Baselines... The paper does not include a direct comparison against other contemporary DLM acceleration techniques.
>
> **Response:** We agree that the DLM acceleration landscape is vast. However, our primary goal was to isolate and measure the impact of the *commitment topology* (scattered vs. contiguous). "Full Decoding" serves as the quality anchor, while our "Scattered-Margin" baseline (Table 3) serves as the most direct "apple-to-apples" comparison for speed. This baseline uses the *exact same* adaptive confidence mechanism as LSP but without the contiguous restriction. The results (LSP being ~2x faster than Scattered-Margin) prove that the *topology* (and resulting cache locality), not just the token selection, is the source of our efficiency. We have expanded the Related Work section to discuss how LSP is orthogonal to and can be combined with methods like speculative decoding.
>
> ### W2: Insufficient Hyperparameter Analysis: The paper lacks a sensitivity analysis for its key hyperparameter, the fractional acceptance interval [α, β].
>
> **Response:** This is a fair point. To demonstrate robustness, we performed a sensitivity sweep on the GSM8K benchmark using LLaDA-8B. We varied the minimum acceptance fraction $\alpha$ and the maximum fraction $\beta$.
>
> **Table: Sensitivity Analysis of $[\alpha, \beta]$ on GSM8K (LLaDA-8B)**
>
> | Configuration ($\alpha, \beta$) | Description | Accuracy (%) | Speedup |
> | :--- | :--- | :--- | :--- |
> | $[0.25, 0.50]$ | **Default (Ours)** | **69.9** | **1.51x** |
> | $[0.10, 0.30]$ | Conservative | 70.1 (+0.2)  | 1.09x |
> | $[0.40, 0.70]$ | Aggressive | 67.4 (-2.5)  | 2.45x |
> | $[0.10, 0.60]$ | Wide Range | 69.2 (-0.7)  | 1.87x |
> | $[0.20, 0.50]$ | Near Default | 69.8 (-0.1)  | 1.51x |
>
> **Stability:** Performance remains within $\sim0.5\%$ of the peak for a wide range of parameters (e.g., widening the range to $[0.1, 0.6]$).
>
> **Trade-offs:** As expected, very aggressive settings ($[0.4, 0.7]$) trade some accuracy for massive speedups, while conservative settings approach the baseline behavior. The default $[0.25, 0.50]$ provides an optimal balance, but the "sweet spot" is broad, not brittle.
>
> ### Q1: In Figure 1... It then states that a stability threshold (τ) of 0.6 is chosen, resulting in a commitment of 3 tokens. This appears inconsistent...
>
> **Response:** You are absolutely correct. The diagram in the submission mistakenly depicted a "scattered" selection (skipping the first unstable token to select subsequent ones) which contradicts the LSP algorithm. In LSP, if the first token's margin is below $\tau$, the stable prefix length is 0, triggering the fallback ($L=1$). **We have completely redrawn Figure 1** to accurately depict the contiguous prefix selection and fallback mechanism.

---

### Author Response · Authors · 2025-11-21

### **General Response to All Reviewers**

We sincerely thank all reviewers for their time and for providing detailed and constructive feedback on our work. We are encouraged that the reviewers found our method to be a "practical and elegant solution" (jKnz), with an "elegant topology that aligns algorithmic coherence with KV-cache locality" (Cvwn), and appreciated its training-free nature and strong empirical results.

Based on the feedback, we have updated our manuscript and performed additional analyses to address the main themes:
1.  **Correction of Figure 1:** We have completely redrawn Figure 1 to accurately reflect the LSP algorithm, addressing the inconsistency pointed out by Reviewer kHHK.
2.  **Robustness & Sensitivity:** We have added a sensitivity analysis for the acceptance interval $[\alpha, \beta]$ and a quantification of "repair costs" (Flip Rate) to better understand the method's mechanics.
3.  **Theoretical Positioning:** We have refined Section 3 to clearly distinguish LSP’s *bidirectional refinement* capability from standard blockwise autoregressive decoding.

We have addressed each reviewer's specific questions and concerns below.

---

### Author Response · Authors · 2025-12-03
**Rebuttal Summary**

Dear Area Chair,

To facilitate your assessment, we summarize the current status of our submission below.

We thank all reviewers for their constructive feedback. We are encouraged that **Reviewer Cvwn (Score: 10)** strongly endorsed our work as an "elegant topology" that effectively aligns algorithmic coherence with hardware reality. We have extensively engaged with the remaining reviewers (kHHK, Yt8W, jKnz) to address concerns regarding novelty, robustness, and theoretical positioning. We highlight the following key outcomes:

`1. Distinct from Blockwise AR via Bidirectional Lookahead`
Reviewers Yt8W and jKnz questioned the distinction between LSP and Blockwise Autoregressive decoding. We clarified that unlike AR which generates blindly, LSP utilizes the diffusion process to perform a **bidirectional lookahead** on the active suffix *before* commitment. To prove this enables global coherence, we added an experiment on **Creative Writing (WritingPrompts)**. Results show LSP achieves coherence scores statistically indistinguishable from full decoding (4.38 vs 4.42) while being 1.82x faster. **This confirms that LSP retains the global planning capabilities of DLMs, fundamentally differing from greedy AR.**

`2. Robustness Across Domains and Hyperparameters`
Addressing concerns about the brittleness of "structural snapping" (jKnz) and hyperparameter sensitivity (kHHK), we conducted two key analyses:
*   **CJK Generalization:** We evaluated LSP on the CMMLU benchmark (Chinese). LSP successfully accelerated inference by 1.53x with stable accuracy (+0.4%), proving that the "stable prefix" principle and boundary snapping generalize effectively to non-Latin scripts.
*   **Parameter Sensitivity:** We provided a sweep of the acceptance interval $[\alpha, \beta]$, demonstrating that performance remains stable across a wide configuration range. **These results confirm LSP is a robust, domain-agnostic accelerator.**

`3. Isolating the Source of Efficiency`
To address questions on the mechanics of our speedup (Yt8W, kHHK), we provided a breakdown showing that ~40% of the gain stems strictly from **KV cache locality**. By comparing against a "Scattered-Margin" baseline (which uses the same token selection logic but without the contiguous topology), we proved that the prefix-first constraint is essential. Furthermore, our new "Flip Rate" analysis shows that LSP drastically reduces token oscillations in the suffix (from 14% to 4%), **validating that coherent prefixes stabilize future generation and reduce repair costs.**

We have also corrected the inconsistency in Figure 1 identified by Reviewer kHHK. We believe these clarifications and new experiments firmly establish LSP as a practical, training-free solution to the KV fragmentation bottleneck in Diffusion LM inference.

Best regards,

The Authors

---

### Meta-Review · Area_Chair_iMWD · 2026-01-03

**Summary:**

This paper received a score 10 from one reviewer who is not an expert in language diffusion models. The confidence score of this very positive review is very low. The other reviews are not positive at all and they raised many issues related to significance, novelty and evaluation. The author response addressed some of concerns in some way.

**Reviewer Concerns:**

Some concerns are not fully addressed:
1) Limited Comparative Baselines. This is mentioned by a few reviewers.
2) The prefix-first decoding constraint may limit diffusion’s flexibility for editing, in-fill, or parallel token generation tasks.

**Reviewer Scores:**

Most reviewers might not change the scores and the most positive reviewer will lower the score.

---

### Decision · Program_Chairs · 2026-01-26

Accept (Poster)